# Formulation of an innovative model for the bioeconomy

C. A. Zuniga-Gonzalez[1]*, J. L. Quiroga-Canaviri[2], J. J. Brambila-Paz[3], S. G. Ceballos-Pérez[4][¤b‡], M. M. Rojas-Rojas[5][¤a‡]

1 Specific Direction of Agroecology, Agricultural and Veterinary Sciences Direction, Research Center in Bioeconomics and Climate Change, National Autonomous University of Nicaragua, Leon, Nicaragua,
2 Postgraduate in Agricultural Economics and Industrial Engineering, Major University of Saint Andrews, La Paz, Bolivia, 3 Postgraduate in Socioeconomics, Statistics and Informatics, Campus Montecillo, Postgraduate College, Texcoco, México, 4 Innovation Research and Postgraduate, Polytechnic University of Pachuca, National Council of Science and Technology, Hidalgo, Mexico, 5 Postgraduate in Agri-Food Science and Technology, Autonomous University of Chapingo, National Council of Science and Technology, Texcoco, Mexico

☙ These authors contributed equally to this work.
¤a Current address: Polytechnic University of Pachuca, State of Mexico
¤b Current address: National Council of Science and Technology, Texcoco, State of Mexico, Mexico
‡ SGCP and MMRR also contributed equally to this work.
* czuniga@ct.unanleon.edu.ni

**Data Availability Statement:** "All relevant data are within the manuscript and its Supporting Information files." "All data files are available from the Mendeley database (DOI 10.17632/kpm9r53srw.1])."

## Abstract

### Background

The bioeconomy, an evolving concept promoting sustainable use of renewable biological resources, confronts the challenge of balancing growth and sustainability across sectors like biotechnology, agriculture, and forestry. This study aims to elucidate the bioeconomy's dynamic nature, constructing a comprehensive theoretical model addressing these complexities.

### Methodology

Through an extensive literature review, foundational elements for this model were identified: defining the core concept, delineating relevant variables, specifying assumptions and parameters, and depicting relationships through equations or diagrams. Special attention was given to integrating Georgescu-Roegen's insights, emphasizing causal links, state variables, measurement scales, and validation plans.

### Results

The model incorporates Georgescu-Roegen's insights, highlighting the importance of clearly defining the bioeconomy for a comprehensive understanding. The proposed model leverages variables, assumptions, and equations within Georgescu-Roegen's framework, serving as a crucial tool for researchers, policymakers, and industry stakeholders. This approach facilitates research structuring, informed decision-making, and interdisciplinary collaboration.

**Funding:** The author(s) received no specific funding for this work.

**Competing interests:** No authors have competing interests.

## Conclusion

By addressing the bioeconomy's evolution, and cross-sectional boundaries, and adopting a broader perspective, this study contributes to policy development for a more sustainable and integrated bioeconomy. Based on empirical knowledge, this model provides not only a solid theoretical framework but also practical guidelines for advancing toward a balanced and resilient bioeconomy.

## 1. Introduction

The bioeconomy is an emerging concept that focuses on the sustainable use of renewable biological resources to shape a more ecologically and economically sustainable future [1]. It is a global trend driven by the need to address resource constraints and advances in microbiology [2]. The bioeconomy encompasses various sectors such as biotechnology, agriculture, and forestry. It offers opportunities for industries and agriculture, including the creation of new jobs and economic opportunities. However, the bioeconomy also poses challenges, such as the need to balance economic growth with environmental sustainability and ensure equitable distribution of benefits [3]. Strategies and policies are being developed to transition to a bioeconomy and enhance sustainability at economic, ecological, and social levels [4, 5].
Understanding the practical implications of the bioeconomy in public policy remains crucial. This study aims to contribute by constructing a theoretical model synthesizing various perspectives within the bioeconomy and providing insights for decision-making processes, particularly in the context of public policy formulation.

The evolving nature of the bioeconomy requires a nuanced understanding of its conceptual evolution and cross-sectional boundaries [6]. By delineating these aspects within the theoretical model, this study aims to provide policymakers with a comprehensive understanding of the bioeconomy's dynamics and facilitate the development of evidence-based policies that promote sustainable development. By developing a theoretical model that elucidates the complexities of the bioeconomy and its implications for decision-making, this study seeks to bridge this gap and provide policymakers with a valuable tool for informed decision-making.

Incorporating Georgescu-Roegen's insights, the model emphasizes the importance of understanding the bioeconomy through causal links, state variables, measurement scales, and validation plans. Georgescu-Roegen's theoretical contributions offer a critical perspective on resource use, entropy, and economic sustainability, which are essential for framing the bioeconomy in a way that addresses both its potential and its limitations.

The objective of this study is to construct a theoretical model that synthesizes and integrates the diverse approaches and perspectives within the field of bioeconomy. This theoretical model will aim to provide a comprehensive framework for understanding the bioeconomy, taking into account its evolving conceptualization, cross-sectional nature, and the shift towards a broader perspective. The model will serve as a tool to enhance our understanding of the bioeconomy, its complexities, and its implications for sustainable development.

## 2. Literature review

The concept of bioeconomy is still evolving and subject to different interpretations, but it encourages fruitful exchange of information and ideas [7, 8].

Overall, the bioeconomy is important for addressing global challenges, promoting sustainable development, and generating innovations for resource utilization and protection [2].

Key insights from the literature include:

1. The concept of the bioeconomy has evolved and been framed differently in different fields and sectors [9, 10].

2. The bioeconomy is seen as a cross-sectional sector that extends across official statistics and cannot be clearly delimited [11].

3. Recent bioeconomy strategies have become more moderate in their promises of economic growth, reflecting a shift towards a broader perspective that considers social-ecological transformation [12].

These insights highlight the need for a theoretical model to clarify the concept's evolution, delineate its boundaries, and guide efforts towards sustainability.

A theoretical model can provide a common reference point for researchers, policymakers, and stakeholders, helping to clarify the concept's evolution and prevent fragmented approaches [7].

Understanding the bioeconomy's cross-sectional nature is essential for mapping its intersections with other sectors. A theoretical model can offer a structured approach to enable a holistic perspective [11].

Incorporating the shift towards sustainability in a theoretical model can guide efforts to align bioeconomic activities with environmental and societal goals [13, 14].

A comprehensive theoretical model can provide a foundation for more effective bioeconomy policies and strategies, balancing economic growth with environmental and social considerations [15, 16].

Synthesizing diverse perspectives can foster interdisciplinary collaboration within the bioeconomy field, facilitating dialogue and knowledge exchange [17, 18].

The theoretical model developed in this study will have broad applicability across academia, policymaking, and industry, serving as a valuable reference for research, policymaking, and strategic planning.

## 2.1 Base concept or theory

The base concept or theory of the bioeconomy centers on the sustainable utilization of renewable biological resources to promote economic growth and ecological sustainability, transitioning from a fossil fuel-based economy to one relying on resources like agricultural products, forestry, and biotechnology. Key elements include (Figs 1 and 2):

The bioeconomy revolves around the use of naturally replenished resources such as crops, forests, and microorganisms, considered sustainable alternatives to finite fossil fuels [19, 20].

Economic Development aims to drive economic development by harnessing the value and potential of biological resources, creating new markets, industries, and jobs [21, 22].

Ecological Sustainability is a fundamental aspect is the commitment to ecological sustainability, seeking to minimize environmental impact, reduce greenhouse gas emissions, and promote responsible land and resource management [23, 24].

Technological Advancements for the bioeconomy relies on advancements in biotechnology, genetic engineering, and other scientific disciplines to enhance resource utilization efficiency, driving innovation and competitiveness [25, 26].

Circular Economy is closely linked to the idea of a circular economy, minimizing waste and by-products, and promoting efficient resource use and recycling [27, 28].

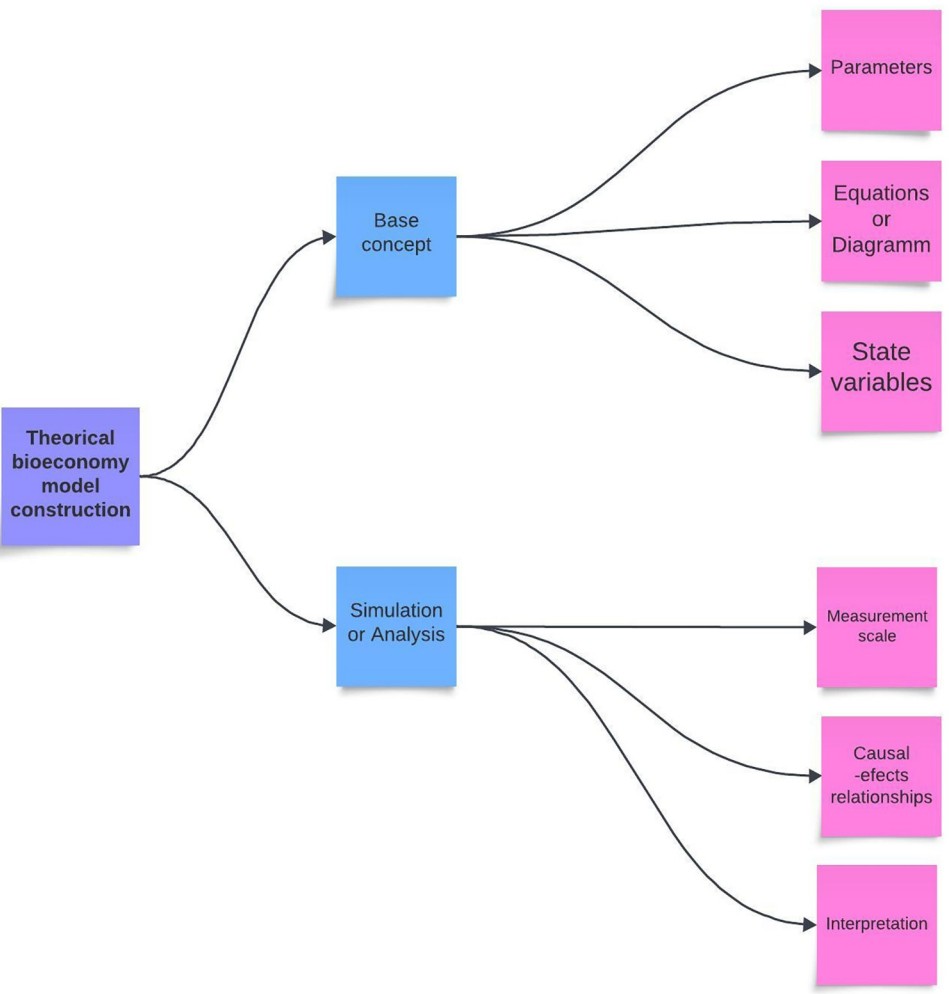

**Fig 1. Key elements of bioeconomy concept.**

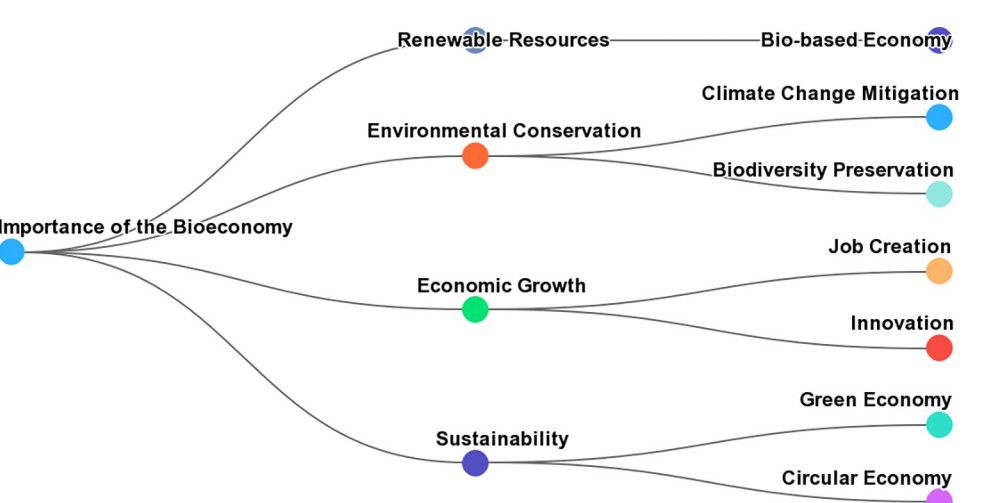

**Fig 2. Conceptual map for the bioeconomy model.**

Cross-Sectoral Span in the bioeconomy spans various sectors including agriculture, energy, and materials, encompassing food and feed production, biofuels, biogas, bioplastics, and bio-materials [29, 30].

Due to reliance on biological resources, ethical considerations are crucial, including issues related to genetic modification, animal husbandry, and equitable distribution of benefits [31, 32].

The base concept underscores the potential to address resource constraints, reduce environmental impact, and create sustainable economic opportunities. However, it is a dynamic and evolving field with various interpretations, approaches, and ongoing discussions about achieving a balance between economic growth and environmental responsibility [19].

In summary, while the bioeconomy, eco-efficiency, and sustainability share the goal of reducing environmental impact and improving efficiency, each approaches this goal from different perspectives and with distinct focuses. DEA is a versatile tool that can be adapted to evaluate efficiency in each of these contexts, providing a quantitative perspective on performance based on resources used and outcomes achieved.

## 2.2 Variables

The selection of variables is informed by the extensive literature on bioeconomic modeling, particularly studies employing Data Envelopment Analysis (DEA) and Stochastic Frontier Analysis (SFA) [33–36].

Chosen variables encompass economic, environmental, and social indicators, reflecting the multidimensional nature of the bioeconomy [37–40].

The rationale for selecting these variables lies in their importance for comprehensively assessing bioeconomic activities' performance and sustainability, aligning with the study's objectives [39, 40].

Causal Relationships and Assumptions:

Causal Relationships: Describe how variables influence each other, whether directly or indirectly, crucial for policymakers, researchers, and stakeholders [41].

Assumptions: Specify the underlying conditions or premises of the model, essential for understanding its limitations and scope [6].

Parameters and Equations/Diagrams:

Parameters: Define constant values affecting relationships between variables, obtained through empirical data, estimations, or assumptions.

Equations/Diagrams: Represent relationships between variables and parameters, facilitating understanding [42].

State Variables and Measurement Scales:

State Variables: Essential for tracking system dynamics over time or space [43].

Measurement Scales: Define qualitative or quantitative scales for variables, crucial for data analysis and interpretation [34, 36, 44].

Simulation/Analysis and Validation:

Simulation/Analysis: Indicate planned methods for exploring the model's behavior, including modeling software or numerical methods [45, 46].

Validation: Provide a plan for comparing model predictions with empirical data, ensuring accuracy and relevance [47, 48].

Interpretation, Sensitivity, and Scenarios:

Interpretation: Describe how model results will be interpreted and their implications [49].

Sensitivity and Scenarios: Conduct sensitivity analysis to evaluate model response to changes, and consider alternative scenarios to explore different conditions and outcomes.

The concept of the bioeconomy (Figs 1 and 2), rooted in the sustainable utilization of renewable biological resources and the promotion of economic growth alongside ecological sustainability, is subject to the influence of various dynamic variables. These variables play a pivotal role in shaping the bioeconomy model and the course it takes. Understanding the intricate interplay between these variables is essential for policymakers, researchers, and stakeholders invested in the bioeconomy [7–10, 50].

Throughout the manuscript, it systematically reference the contributions of various authors who have reevaluated and reimagined the bioeconomy model proposed, thereby enriching its holistic integration [41, 51–53]. These contributions are pivotal in refining our understanding and application of the bioeconomy framework, ensuring its relevance and effectiveness in addressing contemporary challenges.

## 2.3 Economic theories relevant to the bioeconomy

Jiménez [54] provides an overview of economic models, highlighting the evolution of mathematical languages employed in economics from 1838 to the present day. The exposition is structured into three phases, emphasizing the increasing use of linear algebra, game theory, and stochastic processes over time. The article concludes that while economics is a social science, employing the simple language of mathematics facilitates work.

In addition to examining the economic, environmental, and social dimensions of the bioeconomy, this study incorporates advanced analytical techniques such as Data Envelopment Analysis (DEA) and Stochastic Frontier Analysis (SFA) to assess efficiency and productivity. Building upon Georgescu-Roegen's seminal work, these methodologies provide a robust framework for evaluating bioeconomic systems' performance. DEA offers a non-parametric approach to assess relative efficiency, while SFA enables the estimation of frontier functions accounting for random variations and inefficiencies. By integrating these methods into our model, we aim to provide a comprehensive understanding of the bioeconomy's dynamics and its implications for sustainable development [36, 55].

## 2.4 Assumptions of the DEA model—Georgescu-Roegen's bioeconomic model

The proposal of the model requires consideration of Georgescu-Roegen's bioeconomic model within a DEA (Data Envelopment Analysis) and SFA (Stochastic Frontier Analysis) framework, methodologies used in assessing the relative efficiency of productive units. The underlying assumptions in these models can vary, but some common assumptions include [45, 48, 56]:

Homogeneity of Inputs and Outputs: It assumes that inputs and outputs can be uniformly compared across different productive units. This implies that the inputs and outputs are similar in their biomass nature and can be measured and compared equivalently.

Constant Returns to Scale: The technology used for production is considered constant for all units evaluated. This implies that the optimal scale of operation is the same for all units, and there are no efficiency changes as the production scale is altered.

Use of Complete Information: It assumes the availability of complete and accurate information on the inputs and outputs of each productive unit. Lack of complete information may affect the accurate assessment of efficiency.

Absence of Externalities: It assumes that productive units are not influenced by external factors that could affect their efficiency. This implies that the economic environment and other external factors do not impact production or efficiency.

Rational Decision-Making: It assumes that productive units make decisions rationally and seek to maximize their efficiency given the available constraints and resources.

It's essential to note that these assumptions may vary depending on the specific approach within DEA or SFA, and their validity can depend on the particular context of application. Furthermore, the violation of any of these assumptions may affect the validity of the results obtained through these methodologies.

## 2.5 The measurement scopes of DEA-SFA for Georgescu-Roegen's bioeconomic model

Over the past 50 years, various measures have been employed to assess efficiency and technology, generating representations of an efficient frontier. These frontiers have been estimated using two main methods: a) Data Envelopment Analysis (DEA) and b) Stochastic Frontier Analysis, involving mathematical programming and econometric methods, respectively. The most commonly used software includes computer programs such as DEAP and Frontier [44, 57].

DEA involves the application of linear programming methods to construct a non-parametric segment surface or frontier across all the data. Using computer programs, a variety of models are considered, such as:

a. Standard Constant Returns to Scale (CRS) and Variable Returns to Scale (VRS) models that involve the calculation of technical and scale efficiencies [58]

b. The extension of the above models to account for cost and allocative efficiencies [33].

c. The application of Malmquist DEA methods to panel data to calculate indices of total factor productivity (TFP) Change: technological change; technical efficiency change; and scale efficiency change [33].

The SFA with the computer program Frontier version 4.1 provide maximum likelihood estimates of a wide variety of stochastic frontier production and cost functions [34, 35].

## 3 Results

In this section, the results utilized by the bioeconomy with DEA-SFA approaches and the reasons behind it were analyzed. Understanding bioeconomy begins by framing a context, a timeframe, and identifying the issue of scarcity and limitations due to entropy. This assessment allows for the recognition of the interrelationships between the economic process and its environment, as emphasized by Georgescu-Roegen [37].

Georgescu-Roegen's model of bioeconomy is a radical ecological perspective on economics that he developed in the 1970s and 1980s [59].

However, other models of bioeconomy can be compared to Georgescu-Roegen's. One study analyzed three different interpretations of the term bioeconomy, each presenting distinct visions of future economic development and technical trajectories [59].

Another study identified four models for the agricultural non-food bioeconomy, which approach sustainability issues differently [49].

Additionally, the circular bioeconomy (CBE) has been appearing on the political agenda with increasing intensity, and a comparative analysis of CBE strategies in different countries and regions identified common aspects, such as involvement of multiple sectors and policy instruments [47].

These models provide alternative perspectives on the bioeconomy that can be compared to Georgescu-Roegen's, who did not necessarily sympathize with simplifying elements of ceteris paribus, unitary degree of homogeneity and other aspects that have made the function more complex, but whose essence as a theoretical basis for analysis is maintained (Fig 3).

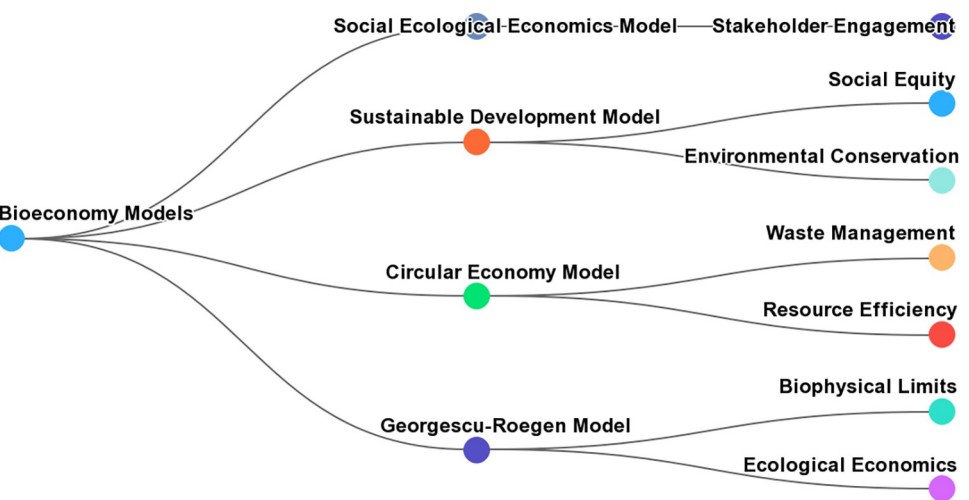

**Fig 3. Alternative bioeconomy models.**

Georgescu-Roegen's bioeconomic model is characterized by several key principles, as outlined in the variables: Methodological Perspective, Flow-Fund Theory, and Alternative to Mainstream Economics (Fig 4).

The methodological perspective from Georgescu-Roegen is criticized the lack of significance of certain economic models and emphasized the limited role of mathematics used in economics [48].

The Flow-Fund Theory from Georgescu-Roegen's flow-fund theory focuses on the relationships between production, the physical dimension of the economic process, and the laws of thermodynamics [45].

The alternative to Mainstream Economics from Georgescu-Roegen's bioeconomic model offers a credible alternative to standard theories of production and emphasizes the study of problems afflicting less developed societies [46, 48].

Overall, Georgescu-Roegen's bioeconomic model incorporates methodological principles, the flow-fund theory, and an alternative perspective to mainstream economics. It highlights the physical dimension of the economic process and the importance of considering the laws of thermodynamics [45, 46, 48] (Fig 4).

As mentioned earlier, Georgescu-Roegen [38] defines two categories: Stock of services S (i) and Flows F(t). This leads to a temporally bounded function in the space from 0 to T, where a series of production processes occur within the range of 0 to T as long as t < T [38–40]. The flow-fund model differentiates between flows (which are consumed or produced in the economic process) and funds (which remain unchanged during the process). Mathematically, this can be expressed as (Eq 1):

$$Q_0(t) = F(t) \cdot L(t) + \sum_i^n R_i(t) \qquad (Eq1)$$

where Q (*t*) represents the output at time (e.g Bio products based on Vegetal Biomass, Animal Biomass, Micro-organisms), F(t) represents the fund factors (e.g., labor, capital), L(t) is the labor input (Stock of services), and $R_i$ (t) are the various flow resources (Flows e.g Renewables Resources (Bio-based Economy: Productive bioeconomy path, Environmental Conservation (Biodiversity conservation, GEI Emission, Economic Growth (Innovation, Bio technology, Job creation, Creative economy, Sustainability (Green Economy, Circular Economy, Inputs

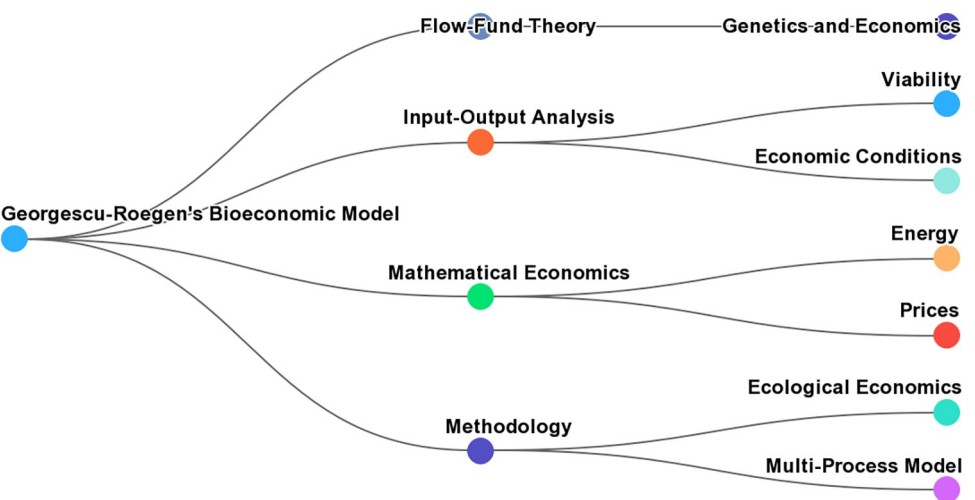

**Fig 4. Key components of Georgescu-Roegen's bioeconomic model.** Source: Scopus AI data.

necessary to keep efficiency intact (Renewable Raw Materials, advanced technology, renewable energy, quality water, skilled human capital, adequated infrastructure, supportive policies and regulations, Capital equipment (infrastructure Research laboratories, efficient production facilities, Product flow (Bio inputs, bio refinery, bioproducts, Ethical consideration Bioethics, ancient culture and Waste product flow as Circular Economy)

The maintenance element present in F (t) is what ensures intact or low-entropy efficiency and, therefore, utility. However, in consumer societies, the issue lies in market competition pressures driving technological change, leading to increased production and high levels of entropy, rendering resource availability non-useful [39].

Georgescu-Roegen emphasized the second law of thermodynamics (entropy law) in economic processes, which can be symbolized as Eq 2:

$$\Delta S = \sum_{i}^{n}\left(\frac{Q_i}{T_i}\right) - \sum_{i}^{n}\left(\frac{Q_j}{T_j}\right) \tag{Eq2}$$

where $\Delta S$ is the change in entropy, and $Q_j$ are heat quantities, and $T_i$ T and $T_j$ are the respective temperatures of the heat exchanges.

## 3.1 The new bioeconomy model

The model development process is grounded in a thorough examination of the existing literature and theoretical frameworks within the field of Bioeconomy. Drawing upon this foundation, it meticulously formulated the model through a systematic process that involved the identification of key variables, the integration of relevant theories, and the adoption of appropriate methodological approaches. Throughout this process, it remained cognizant of the complexities inherent in developing a comprehensive model that adequately captures the dynamics of Bioeconomy. Furthermore, it rigorously validated and evaluated our model through DEA-SFA, ensuring its robustness and reliability. By contextualizing the development of the model within the broader framework of existing research and methodological considerations, it aim to provide readers with a clearer understanding of its significance and relevance within the bioeconomy.

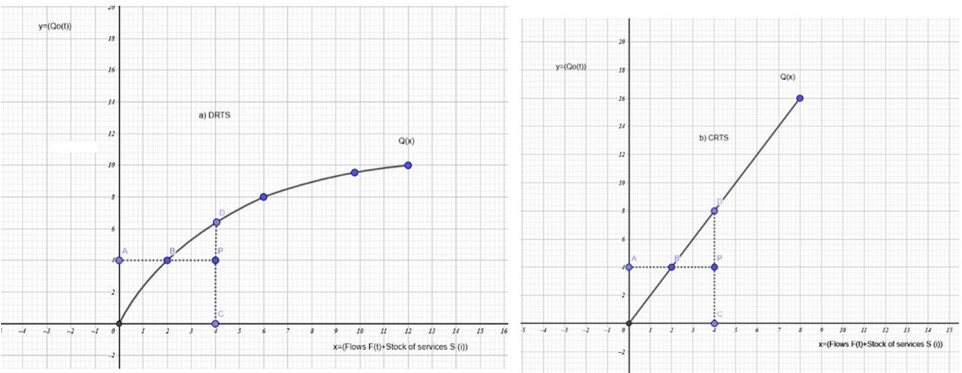

**Fig 5. Illustration of the Georgescu model applying the DEA methodology Input- and output-orientated technical efficiency measures and returns to scale.**

DEA is a methodology applied across industries to gauge efficiency, while the bioeconomy emphasizes sustainable utilization of biological resources. While "DEA bioeconomy" isn't explicitly discussed, comprehending DEA and the bioeconomy independently unveils potential intersections and applications between the two realms.

Fig 5 illustrates the case of the Georgescu model applying the DEA methodology with the CCR-CRS and BCC-VRS input-output oriented methods. It is noted that the inputs are represented in the model by the Flows F(t), which are composed of variables such as R, EC, EG, S, I, M, ET, and W, in addition to the group of variables used in traditional economic models such as L, K, and H. The y-axis is defined by the bio-products such as bio-inputs, biorefineries, etc.

Fig 5(A) shows where it have a decreasing returns to scale technology represented by Q(x), and an inefficient firm operating at the point P. The Farrell input-orientated measure of TE would be equal to the ratio AB/AP, while the output-orientated measure of TE would be CP/CD. The output- and input-orientated measures will only provide equivalent measures of technical efficiency when constant returns to scale exist, but will be unequal when increasing or decreasing returns to scale are present by Fare and Lovell [60]. The constant returns to scale case is depicted in Fig 5(B) where it observe that AB/AP = CP/CD, for any inefficient point P it care to choose.

**3.1.1 DEA- SFA Georgescu-Roegen's bioeconomic model.** To capture the efficiency and productivity within the bioeconomy, production functions are used. These functions must consider the constraints imposed by biological and physical laws. A standard production function in this context can be represented as Eq 3:

$$Q = A \cdot K^{\alpha} \cdot L^{\beta} \cdot M^{\gamma} \tag{Eq3}$$

where Q is the output, A is a technological constant, K is the capital input,

L is the labor input, M is the material input, and α, β, γ are the output elasticities of each input.

Data Envelopment Analysis (DEA) is utilized to measure the relative efficiency of decision-making units (DMUs). The basic DEA model can be written as Eq 4:

$$Maximize\ \theta = \sum_{i}^{n} (u_r y_{rj})$$

subject to:

$$\sum_{i=1}^{m}(u_r y_{rj}) = 1$$

$$\sum_{r=1}^{s}(u_r y_{rk}) - \sum_{i=1}^{m}(u_i y_{ik}) \leq 0, \ \forall k \tag{Eq4}$$

where θ is the efficiency score, $u_{r\_}$ and $v_i$ are weights, $y_{rj}$ is the output, and $x_{ij}$ is the input.

Stochastic Frontier Analysis (SFA) models account for inefficiency and random error. The basic form of the stochastic frontier model is Eq 5:

$$y_i = f(x_i; \boldsymbol{\beta}) + v_i - u_i \tag{Eq5}$$

where $y_i$ is the output, $f(x_i;\beta)$ is the production function, $v_i$ is the random error, and $u_i$ is the inefficiency term.

*3.1.1.1 The Constant Returns to Scale model CCR-(CRS).* The model CCR-CRS (Constant Returns to Scale) proposed by the paper by Charnes, Cooper and Rhodes [61] assumption in Data Envelopment Analysis (DEA) is only appropriate when all Decision Making Units (DMUs) are operating at an optimal scale. For instance, consider a set of farms producing certain crops. Assuming CRS would mean presuming that all these farms are operating precisely at the optimal scale for that particular production. If a farm operates below its optimal scale, the CRS assumption might not hold as some farms could have idle capacity or might not be maximizing their efficiency at that specific scale of production. Assume there is data on K inputs and M outputs on each of N Firms or Decision Making Unit (DMU). For the i-th DMU these are represented by the vectors $F_i S_i$ and $Q_i$ respectively, For the Georgescu-Roegen's Bioeconomic Model $F_i S_i$ or K inputs consider Flows (F) and Stock of services (S) and M outputs consider the $Q_0$: Bioeconomy Output (Vegetal Biomass, Animal Biomass, Micro-organisms). The K x N input matrix, F S, and the M x N output matrix, Q, represent the data of all N DMU's. The purpose of DEA- Georgescu-Roegen's bioeconomic model is to construct a non-parametric envelopment frontier. The best way to introduce DEA is via ratio form. The LP or Eq 1 involves finding the values for μ and υ, such that the efficiency measure of the i-th DMU is maximized, subject to the constraint that all efficiency measure must to be less than or equal to one Eq 6.

$$max_{\mu,\upsilon}(\mu' q_i),$$

$$st. \quad \upsilon' f_i s_i = 1,$$

$$\mu' q_j + \upsilon' f_j \leq 0, \ J = 1, 2, 3, \ldots .N$$

$$\mu, \upsilon \geq 0, \tag{Eq6}$$

Where the notation $\mu$ and $\upsilon$ reflect the multiplier form of the linear programming problem. The $\mu$ and $\upsilon$ weights can be interpreted as normalized shadow prices [43]. Using the duality in

linear programming, one can derive an equivalent envelopment form of this problem as Eq 7.

$$min_{\theta,\lambda}(\theta),$$

$$st. \quad -q_i + Q\lambda \geq 0,$$

$$\theta f_i \, s_i - FS\lambda \geq 0,$$

$$\lambda \geq 0, \tag{7}$$

Where $\theta$ is a scalar and $\lambda$ is a Nx1 vector of constants. This envelopment form involves fewer constraints than the multiplier form ((K+M < N+1), and hence is generally the preferred form to solve. The value of $\theta$ obtained will be the efficiency score for the i-th DMU. It will satisfy $\theta \leq 1$, with a value of 1 indicating a point on the frontier and hence a technically efficient DMU in the sample. A value of $\theta$ is then obtained for each DMU. Farrell [43] introduce the input and output slack ($\lambda$), for i-th DMU the output slacks will be equal to zero only if $Q\lambda - q_i = 0$, while the input slack will be zero only if $\theta f_i s_i - FS\lambda = 0$ (for the given optima values of $\theta$ and $\lambda$. The DEAP software gives the user three choices regarding the treatment of slacks: One-stage DEA, Two-stage DEA, and mult-satge DEA.

*3.1.1.2 The Variables Returns to Scale model (VRS) and scale efficiencies.* Imperfect market conditions like imperfect competition or financial constraints can lead a Decision Making Unit (DMU) to operate below its optimal scale. In their work, Banker, Charnes, and Cooper [62] proposed an enhancement to the DEA model (BCC model) to address variable returns to scale (VRS) scenarios. Using the VRS approach allows for the computation of Technical Efficiency (TE) without these Scale Efficiency (SE) influences. Utilizing the CRS specification in situations where not all DMUs are operating at their optimal scale could yield TE metrics entangled with scale efficiencies (SE). Input and Output Orientations.

The CRS linear programming problem can be easily modified to account for VRS by adding the convexity constrain: $N1'\lambda = 1$ to Eq 8.

$$min_{\theta,\lambda}(\theta),$$

$$st. \quad -q_i + Q\lambda \geq 0,$$

$$\theta f_i \, s_i - FS\lambda \geq 0,$$

$$N1'\lambda = 1$$

$$\lambda \geq 0, \tag{8}$$

Where N1 represent an Nx1 vector filled with ones. This methods constructs a convex hull intersecting planes that enclose the data point more approach forms a convex hull of intersecting planes which envelope the data points more tightly compared to the CRS conical hull. As result, it generates technical efficiency scores that are either equal to or greater than those derived from the CRS model. During the 1990's the VRS specification became widely used. However, one limitation of this scale efficiency measure is its inability to determine whether a DMU operates under increasing or decreasing returns to scale. To address this, an additional DEA problem can be conducted by imposing non-increasing returns to scale (NIRS). This adjustment involves modifying Eq 3 in the DEA model by substituting the $N1'\lambda = 1$ restriction

with $N1'\lambda \leq 1$, to provide Eq 9.

$$min_{\theta,\lambda}(\theta),$$

$$st. \quad -q_i + Q\lambda \geq 0,$$

$$\theta f_i s_i - FS\lambda \geq 0,$$

$$N1'\lambda \leq 1$$

$$\lambda \geq 0, \tag{9}$$

*3.1.1.3 Input and output orientations.* The output-orientated models are very similar to their input-orientated counterparts. Consider the example of the following out-orientated VRS model Eq 10.

$$max_{\phi,\lambda}(\phi),$$

$$st. \quad -\phi q_i + Q\lambda \geq 0,$$

$$f_i s_i - FS\lambda \geq 0,$$

$$N1'\lambda = 1$$

$$\lambda \geq 0, \tag{10}$$

Where $1 \leq \phi < \infty$, and $\phi - 1$ is the proportional increase in outputs that could be achieved by th i-th DMU, with input quantities held constant. Note that $^1/_\phi$ defines a TE score which varies between zero and one (and that this is the output-orientated TE score reported by the software DEAP.

*3.1.1.4 Prices information and allocative efficiency (Eq 11).*

$$min_{\lambda,f_i^*}(w_i'F_iS_i^*),$$

$$st. \quad -q_i + Q\lambda \geq 0,$$

$$f_i s_i^* - FS\lambda \geq 0,$$

$$N1'\lambda = 1$$

$$\lambda \geq 0, \tag{11}$$

Where $w_i$ is the vector of input prices for the i-th DMU and $F_i s_i^*$ (which is calculated by the LP) is the cost-minimizing vector of input quantities for the i-th DMU, given the input prices $w_i$ and the output levels $Q_i$ The total cost efficiency (CE) or economic efficiency of the i = th DMU would be calculated as $CE = {}_i^w {}'f_i s_i^* / {}_w i' f_i s_i$ That is, the ratio of minimum cost to observed cost. One can calculate the allocative efficiency (AE) residually as $AE = CE/TE$ [49]

*3.1.1.5 Panel data, DEA and the Malmquist index.* Färe *et al.* [58] proposed an output-based Malmquist productivity change and technical efficiency change Eq 12.

$$\psi_0\left(\varrho_{t+1}, f_{t+1}, s_{t+1}, \varrho_t, f_t\right) = \left[\frac{\delta_0^t(f_{t+1}, s_{t+1}, \varrho_{t+1})}{\delta_0^t(f_t, s_t, Q_t)} \times \frac{\delta_0^{t+1}(f_{t+1}, s_{t+1}, \varrho_{t+1})}{\delta_0^{t+1}(f_t, s_t, \varrho_t)}\right]^{\frac{1}{2}} \tag{12}$$

LP$_8$ represents the productivity at the point $(f_{t+1}s_{t+1}, \varrho_{t+1})$ relative to the production point $(fs, \varrho_t)$. In such a way that we consider values from 0 to 1. It will understand that there is a growth of the TFP when the value is 1 from period t to period t+1. To estimate LP $_8$, the four functions of the distances of the components must be calculated, of which the LP problems are involved (similar to those conducted to calculate the Farrel [43], measure in technical efficiency (TE).

Assuming CRS technology. The oriented PL CRS output used to compute $\delta_0^t(f_t s_t, \varrho_t)$ is defined in LP $_8$, $_9$, however, the constraint on convexity (VRS) has been removed and the subscription time is included. This is Eq 13:

$$[\delta_0^t(f_t\ s, \varrho_t)]^{-1} = max_{\phi,\lambda}\phi,$$

$$s.t\ -\phi f_{it}\ S_{it} + Q_t\lambda \geq 0,$$

$$f_{it}s_{it} - F_t S_t\lambda \geq 0,$$

$$\lambda \geq 0, \tag{13}$$

The LP problems are a simple variation of this Eqs 14,15, 16:

$$[\delta_0^t(f_t\ S_t, \varrho_t)]^{-1} = max_{\phi,\lambda}\phi,$$

$$s.t\ -\phi\varrho_{i,t+1} + F_{t+1}, S_{t+1}\lambda \geq 0,$$

$$f_{i,t+1}, s_{i,t+1} - F_{t+1}, S_{t+1}\lambda \geq 0,$$

$$\lambda \geq 0, \tag{14}$$

$$[\delta_0^t(F_{t+1}, S_{t+1}, \varrho_{t+1})]^{-1} = max_{\phi,\lambda}\phi,$$

$$s.a\ -\phi Q_{i,t+1} + F_t S_t\lambda \geq 0,$$

$$f_{it} - F_t S_t \lambda \geq 0,$$

$$\lambda \geq 0, \tag{15}$$

$$[\delta_0^{t+1}(f_t S_t, \varrho_t]^{-1} = max_{\phi,\lambda} \phi,$$

$$s.a \ - \phi \varrho_{it} + F_t S_t \lambda \geq 0,$$

$$f_{it} - F_{t+1}, S_{t+1} \lambda \geq 0,$$

$$\lambda \geq 0, \tag{16}$$

Note that LP $_{11, 12}$ production points are compared with different period-type technologies, the parameter $\phi$ does not need to be $\geq 1$, as when calculating the Farrell [43] efficiency. The points must be below the production amount allowed.

This is most likely to occur at LP $_{10}$, where the period t 1 production point is associated with the technology with period t. With technological advances, values of $\phi < 1$ are possible. Note that if a tech comeback happens, it could happen in LV 10 as well, but it's unlikely.

A few things to notice are that $\phi$, and $\lambda$ are likely to take different values within 4 of LP. Also, note that all four PLs must be computed for each region in the sample. Also note that if you add a period, you will need to calculate 3 LPs per region (to create the correction rate). If we are measuring the T period, we need to calculate the (3T-2) PL for each region in the sample. So for N Region = 6, we need to compute N * (3T-2) LP. This study with N = 6 regions and T = 10 time periods (2012–2021) should provide 6 * (3*10–2) = 168 LPs.

**3.1.2 SFA- Georgescu-Roegen's bioeconomic model.** The stochastic frontier production function was independently proposed by Aigner, Lovell and Schmidt [63] and Meesusen and van den Broeck [42]. The original specification involved a production function specified for cross-sectional data which had an error term which had two components, one to account for random effects and another to account for technical inefficient. This model can be expressed in Eq 17.

$$Q_i = f_i \beta + S_i \beta + (V_i - U_i), \quad i = 1, 2, \ldots \ldots \ldots N \tag{17}$$

Where $Q_i$ is the bioeconomy production (Vegetal Biomass, Animal Biomass, Micro-organisms) (or the logarithm of the bioeconomy production) of the i-th DMU;

$f_i$ is a kx1 vector of (transformation of the) resources input quantities of the i-th firm;

$S_i$ is a kx1 vector of (service of the) the benefits provided by ecosystems and natural resources to the economy and society without being transformed into material goods. These services include things like air and water purification, crop pollination, climate regulation, among others. They are essential for the functioning of the economy but aren't always directly accounted for in traditional economic models.

$\beta$ is a vector of unknown parameters;

The $V_i$, are random variables which are assumed to be independently and identically distributed (iid). $N(0, \sigma_v^2)$, and independent of the $\mu_i$ which are non-negative random variables which are assumed to account for technical inefficiency in bioeconomy production and are often assumed to be iid. $\lceil N(0, \sigma_u^2), \rceil$

*3.1.2.1 The Battese and Coelli [42] specifications.* Battese and Coelli [33] propose a stochastic frontier production function for (unbalanced) panel data which has firm effects which are

assumed to be distributed as truncated normal random variables, which are also permitted to vary systematically with time Eq 18.

$$Q_{it} = f_{it}\beta + S_{it}\beta + (V_{it} - U_{it}), \quad i = 1, 2, \ldots\ldots\ldots N; t = 1, 2, 3, \ldots\ldots\ldots\ldots T \quad (18)$$

Where $Q_{it}$ is the bioeconomy production (Vegetal Biomass, Animal Biomass, Micro-organisms) (or the logarithm of the bioeconomy production) of the i-th DMU in the t-th period; $f_{it}$ is a kx1 vector of (transformation of the) resources input quantities of the i-th firm in the t-th period; $S_{it}$ is a kx1 vector of (service of the) the benefits provided by ecosystems and natural resources to the economy and society without being transformed into material goods. These services include things like air and water purification, crop pollination, climate regulation, among others. They are essential for the functioning of the economy but aren't always directly accounted for in traditional economic models in the t-th period.

$\beta$ is as defined earlier;

The $V_{it}$, are random variables which are assumed to be independently and identically distributed (iid). $N(0, \sigma_v^2)$, and independent of the

$\mu_i(\mu_i exp(-\eta(t - T)))$, where $\mu_i$ are non-negative random variables which are assumed to account for technical inefficiency in bioeconomy production and are often assumed to be iid. $\lceil N(0, \sigma_u^2), \rceil$, distribution; $\eta$ is a parameter to be estimated;

## 3.2 The Battese and Coelli [43] specifications

Kumbhakar, Ghosh and McGukin [64] and Reifschneider and Stevenson [65] propose stochastic frontier models in which the inefficiency effects are expressed as an explicit function of a vector of firm-specific variables and random error. Battese and Coelli [35] propose a model which is equivalent to the Kumbhakar, Ghosh and McGukin [64] specification, with the exceptions that allocative efficiency is imposed, the first-order profit maximizing conditions removed, and panel data is permitted Eq 19.

$$Q_{it} = f_{it}\beta + (V_{it}U_{it}), \quad i = 1, 2, 3 \ldots\ldots\ldots\ldots\ldots\ldots, N, \quad t = 1, 2, 3 \ldots\ldots\ldots\ldots, T \quad (19)$$

Where $Q_{it} f_{it}$ and $\beta$ are defined earlier;

The $V_{it}$, are random variables which are assumed to be iid. $N(0, \sigma_v^2)$, and independent of the $U_{it}$, which are non-negative random variables which are assumed to account for technical inefficiency in bioeconomy production and are often assumed to be iid. $\lceil N(0, \sigma_u^2), \rceil$; Where Eq 20 is:

$$m_{it} = Z_{it}\delta \quad (20)$$

Where $Z_{it}$ is a px1 vector of bio variables which may influence the efficiency of a firm; and $\delta$ is a 1xp vector of parameter to be estimated.

## 3.3 Bioeconomy model (Eq 21)

DEA is a technique used to measure efficiency in various industries, while the bioeconomy focuses on the sustainable use of biological resources. Although there is no direct mention of "DEA bioeconomy," understanding DEA and the bioeconomy separately can provide insights into their potential intersection and applications [6, 65–67]. This model is based in Georgescu-Roegen's Bioeconomic Model. There is limited research on the application of Data Envelopment Analysis (DEA) and Stochastic Frontier Analysis (SFA) in the context of bioeconomy. Most academic contributions to the field of bioeconomy focus on science perspectives, such as

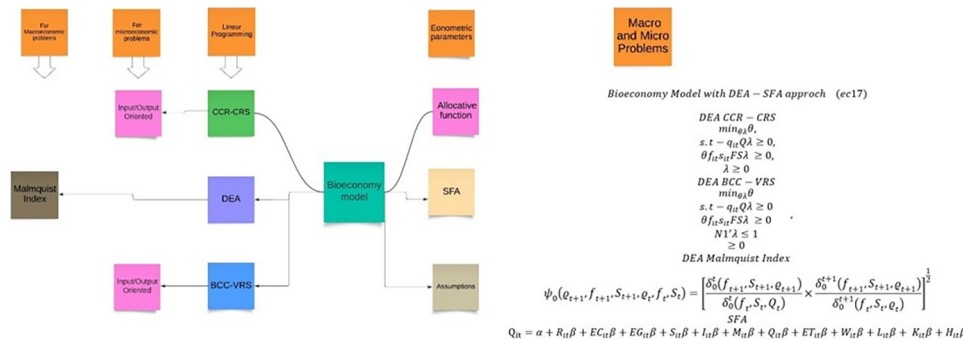

**Fig 6. Georgescu-Roegen's bioeconomic model integrating DEA and SFA: Applications and perspectives.**

chemistry, engineering, and biomedicine, rather than social science perspectives [41] (Fig 6).

$$\textit{Bioeconomy Model with DEA} - \textit{SFA approch}$$

$$\textit{DEA CCR} - \textit{CRS}$$

$$min_{\theta\lambda}\theta,$$

$$s.t - q_{it}Q\lambda \geq 0,$$

$$\theta f_{it}s_{it}FS\lambda \geq 0,$$

$$\lambda \geq 0$$

$$\textit{DEA BCC} - \textit{VRS}$$

$$min_{\theta\lambda}\theta$$

$$s.t - q_{it}Q\lambda \geq 0$$

$$\theta f_{it}s_{it}FS\lambda \geq 0 \qquad , \text{(Eq21)}$$

$$N1'\lambda \leq 1$$

$$\geq 0$$

$$\textit{DEA Malmquist Index}$$

$$\psi_0\left(\varrho_{t+1}, f_{t+1}, S_{t+1}, \varrho_t, f_t, S_t\right) = \left[\frac{\delta_0^t(f_{t+1}, S_{t+1}, \varrho_{t+1})}{\delta_0^t(f_t, S_t, Q_t)} \times \frac{\delta_0^{t+1}(f_{t+1}, S_{t+1}, \varrho_{t+1})}{\delta_0^{t+1}(f_t, S_t, \varrho_t)}\right]^{\frac{1}{2}}$$

$$\textit{SFA}$$

$$Q_{it} = \alpha + R_{it}\beta + EC_{it}\beta + EG_{it}\beta + S_{it}\beta + I_{it}\beta + M_{it}\beta + Q_{it}\beta + ET_{it}\beta + W_{it}\beta + L_{it}\beta + K_{it}\beta + H_{it}\beta$$

## 4 Discussion

DEA (Data Envelopment Analysis) and SFA (Stochastic Frontier Analysis) present promising avenues for optimizing resource allocation in the bioeconomy. The integration of these methodologies into our newly constructed model offers innovative approaches with practical implications [68, 69].

DEA serves as a powerful tool to uncover inefficiencies and productivity gaps within various sectors, including the bioeconomy [70]. By quantifying the utilization of bioresources, DEA enables the assessment of resource efficiency in enterprises, considering both economic and environmental factors in resource allocation [56, 62, 71].

SFA complements DEA by identifying profit drivers and providing flexible strategies for resource allocation [53, 72]. This methodology offers a nuanced approach to eco-efficiency assessment, such as evaluating carbon footprint and pinpointing inefficiency sources within the bioeconomy [52, 63].

The discussion further examines various bioeconomy models, juxtaposing them with Georgescu-Roegen's foundational model. It elucidates fundamental differences in methodological focus, flow-fund theory, and alternatives to mainstream economics, highlighting the advantages and limitations of each model in diverse economic and environmental contexts [34, 36, 43].

Georgescu-Roegen's model underscores ethical and environmental considerations, emphasizing biodiversity conservation, greenhouse gas emissions reduction [38–40], and the promotion of a circular economy [30, 31]. Integrating these aspects into the analysis of efficiency and productivity using DEA and SFA offers a holistic approach to sustainable resource management [10, 16, 73].

Despite progress, challenges persist in transitioning to a more sustainable bioeconomy, including resistance to structural changes and coordination deficits across sectors. The discussion explores strategies to overcome these challenges, emphasizing the role of effective public policies in fostering a resilient and equitable bioeconomy [52, 53, 56, 73].

Overall, the integration of DEA and SFA methodologies into our model represents a significant step forward in enhancing resource efficiency, profitability, and eco-efficiency within the bioeconomy. By addressing inefficiencies and optimizing resource allocation, our model contributes to the advancement of sustainable economic practices in harmony with environmental conservation [74].

The discussion addresses the social impact of bioeconomic strategies, highlighting their potential to create jobs, improve livelihoods, and foster inclusive growth. Strategies to enhance social equity and community resilience through effective public policies are also explored, recognizing the role of governance in shaping equitable outcomes [73, 75–77].

## 5 Concluding remarks

With the proposed model, valuable insights for decision-makers at various levels can be gleaned, facilitating analysis at regional, national, and production unit levels. This includes assessing productivity and informing public agendas.

Analyzing various bioeconomy models, including Georgescu-Roegen's radical ecological perspective, underscores the diversity of approaches available to tackle future economic and sustainability challenges. Policymakers can leverage these insights to consider alternative frameworks that integrate environmental concerns into economic decision-making processes. Acknowledging the interplay between economic processes and environmental constraints can aid in formulating strategies that promote sustainable development and resilience amidst global environmental challenges.

Additionally, the integration of Data Envelopment Analysis (DEA) and Stochastic Frontier Analysis (SFA) methodologies within bioeconomic modeling provides practical tools for measuring efficiency and productivity. Despite limited research in this area, integrating DEA and SFA enhances understanding of resource utilization and sustainable development pathways. This analytical approach empowers decision-makers with actionable insights to identify inefficiencies and optimize resource allocation within the bioeconomy, thus enabling evidence-based policies and initiatives that foster sustainable economic growth while mitigating environmental degradation.

It is notable that while there is limited research on the application of DEA and SFA in the bioeconomy context, significant contributions have been made in various areas. These include

addressing methodological and variable-related challenges in assessing the social, economic, and environmental impacts of the bioeconomy, outlining policy recommendations for bioeconomic growth and global leadership, and emphasizing the importance of stakeholder involvement in transitioning to a circular bioeconomy [51].

## Supporting information

**S1 File. Theoretical example of DEA application in bioeconomy.**
(PDF)

**S2 File. Purpose of mathematical models in bioeconomy.**
(PDF)

## Author Contributions

**Conceptualization:** C. A. Zuniga-Gonzalez, J. J. Brambila-Paz, S. G. Ceballos-Pérez, M. M. Rojas-Rojas.

**Data curation:** M. M. Rojas-Rojas.

**Formal analysis:** C. A. Zuniga-Gonzalez, J. L. Quiroga-Canaviri, J. J. Brambila-Paz.

**Investigation:** C. A. Zuniga-Gonzalez, J. L. Quiroga-Canaviri, S. G. Ceballos-Pérez, M. M. Rojas-Rojas.

**Methodology:** C. A. Zuniga-Gonzalez, J. L. Quiroga-Canaviri, J. J. Brambila-Paz.

**Resources:** S. G. Ceballos-Pérez, M. M. Rojas-Rojas.

**Supervision:** C. A. Zuniga-Gonzalez, J. L. Quiroga-Canaviri.

**Validation:** C. A. Zuniga-Gonzalez.

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
