## [Decision Letter · Decision Letter 0]

31 Jan 2024

PONE-D-23-43140Formulation of an Innovative Model for the Bioeconomy: Unraveling the Secrets of a Sustainable FuturePLOS ONE

Dear Dr. Zúniga-González,

Thank you for submitting your manuscript to PLOS ONE. After careful consideration, we feel that it has merit but does not fully meet PLOS ONE’s publication criteria as it currently stands. Therefore, we invite you to submit a revised version of the manuscript that addresses the points raised during the review process.

We look forward to receiving your revised manuscript.

Kind regards,

Noé Aguilar-Rivera

Academic Editor

PLOS ONE

Journal Requirements:

"No authors have competing interests."

Reviewers' comments:

Reviewer's Responses to Questions

**Comments to the Author**

1. Is the manuscript technically sound, and do the data support the conclusions?

Reviewer #1: No

Reviewer #2: No

Reviewer #3: No

2. Has the statistical analysis been performed appropriately and rigorously? 

Reviewer #1: N/A

Reviewer #2: No

Reviewer #3: No

3. Have the authors made all data underlying the findings in their manuscript fully available?

Reviewer #1: No

Reviewer #2: No

Reviewer #3: No

4. Is the manuscript presented in an intelligible fashion and written in standard English?

Reviewer #1: No

Reviewer #2: No

Reviewer #3: No

5. Review Comments to the Author

Reviewer #1: I appreciate the possibility of reading this paper. It focuses on a topic increasingly relevant in academia and public policy.

I strongly value the mathematical development presented by the authors. The paper displays mastery of the theoretical application of DEA and SFA and how they can be used to evaluate options and make environmentally sustainable decisions.

Nonetheless, the paper is written in an unusual way for academic writing, making its readiness difficult. For instance, the introduction states the paper's goal but doesn't establish a context where this decision-making modeling can work. Later in the paper, one possible deduction is that the authors are thinking in public policy, but this is left to the readers to speculate.

Also, in the introduction, some bullet ideas mention critical elements from the literature review, but these elements are not further developed in the paper. Later, there is a list of the paper's contributions; however, they are not linked with other sections of the introduction or literature about the topic.

Literature review: This section requires a profound review. Some constitutive elements of the concept are presented, but this is not a literature review. Here, as a reader, I expect to get information about the recent literature on bioeconomy, its applications, theoretical models, measurements, and so forth. In other words, I would like to read a review that allows me to understand the state of knowledge in this field and how this paper fits into the current literature. Additionally, due to the declaration of importance of this theory, a comprehensive explanation of Georgescu-Roegen's framework is deeply missed.

Variables: This section presents methodological elements that must be considered when conducting such an endeavor, but it does not explain the chosen variables and the theoretical intuitions behind them.

The following sections are challenging to follow up because it is not clear its intention to develop the paper's central argument. For example, the section "Economic theories relevant to the bioeconomy" is based on one author, and it presents very well-known ideas regarding the evolution of mathematical tools in economics. It is not clear why this information would be important in this context.

Finally, I strongly suggest a Discussion section. It is necessary to explain what these findings mean and how they contribute to the most recent literature. Here, I find the lack of context for how this model was developed particularly complex. It is unclear if this model was thought for companies, countries, regions, or other subjects. Furthermore, an exploration of the model's limitations is required.

Reviewer #2: This study aims to elucidate the bioeconomy's dynamic nature, constructing a comprehensive theoretical

model.

It is quite interesting, with well supported data. However, there are quite many questions about the study.

Reviewer #3: This is a confusing paper describing production functions and approaches to estimate productivity, claiming that would be a model of the Bioeconomy. It took me to read until the very end of the paper when the authors finally describe why they have presented all these equations. Most sections are completely unrelated to each other. Page numbers are missing (neither is there line numbering), making it really difficult to review. No numbering of sections, many headers the same size and font. Unclear how the paper is structured. Zero innovative content, not a scientific paper.

More detailed comments:

Introduction

- Strange spacing, paragraphs after single sentences

- Strange structure, paragraphs starting with bullet points

- Typo: “the review literature“

- The following paragraph is a complete repetition of the previous sentences and redundant: “In conclusion, the bioeconomy does not have a single model, but rather encompasses diverse approaches and perspectives. It is important to consider the evolution of the concept, the crosssectional nature of the sector, and the shift towards a broader perspective for a comprehensive understanding of the bioeconomy [7, 8 9].”

- More one to one repetitions: “The bioeconomy has been framed differently across various fields and sectors”; followed by more repetitions of the exact same sentences

- The introduction is a repetition of the same arguments and the same references over and over again, using almost the exact same words, completely blown up without much content.

Literature review

- Figures 1 and 2 are mixed up. The first part of the lit review seems to talk about figure 2 although there is no 1-to-1 overlap of text and figure. Both figures have no real sources. Figure 1 states the source: Scopus AI data. Unclear what that is.

- Shortest literature review ever

Variables

- Under the header “variables” comes a definition of different parts of models. There are no sources provided and it looks like it comes directly out of ChatGPT, because it is written by directly addressing the reader and giving a cooking recipe.

- Economic Theories Relevant to the Bioeconomy: Suddenly change of citation style and strange paragraph on the history of economic models based on only one source

- Assumptions of the DEA Model - Georgescu-Roegen's Bioeconomic Model: unclear listing of assumptions of DEA (Data Envelopment Analysis) and SFA (Stochastic Frontier Analysis) without the provision of sources

- The measurement scopes of DEA-SFA for Georgescu-Roegen's bioeconomic model: now comes a definition of DEA (Data Envelopment Analysis) and SFA (Stochastic Frontier Analysis); apparently there should some productivity measurement, but I only try to read between the lines

- Results and discussion: strange title, completely unrelated to text, talks about Georgescu-Roegen's model of Bioeconomy, but means “bioeconomics”; some other models and concepts are mentioned without explaining them

- Again figures 3 and 4 with source: Scopus AI data. Unclear what that is. Figures not really explained, but also seem trivial

- A function is shown for Georgescu-Roegen’s Bioeconomic Model but not really explained, sources are in Spanish

- Then comes a description of The Constant Returns to Scale Model (CRS), DEA, some linear programming but it is all completely unclear how that relates to a comprehensive model for the bioeconomy

- 1.5) Panel Data, DEA and the Malmquist Index: more ways to measure productivity

- SFA- Georgescu-Roegen's bioeconomic model: explaining a production function

- “They are essential for the functioning of the economy but aren't always directly accounted for in traditional economic models.” Exact same sentence twice

- Some other production function approaches are listed

- Bioeconomy model (Equation 17): this is the most interesting section in the whole paper because it finally explains “DEA (Data Envelopment Analysis) and SFA (Stochastic Frontier Analysis) have potential applications in optimizing resource allocation in the bioeconomy.” And what the use of all the presentation of production function was. But there is no application at all. Just some arguments from the literature.

Concluding remarks

- Completely unrelated to the whole paper.

6. PLOS authors have the option to publish the peer review history of their article (what does this mean?). If published, this will include your full peer review and any attached files.

Reviewer #1: No

Reviewer #2: No

Reviewer #3: No

---

## [Author Response · Author response to Decision Letter 0]

9 Apr 2024

Noé Aguilar-Rivera

Editor Academico

Dear editor, thank you for all your observations that surely contribute to improving the quality of the research. Below I present in detail each of the improvements incorporated.

Reviewer # 1

[1] Reviewer #1: I appreciate the possibility of reading this paper. It focuses on a topic increasingly relevant in academia and public policy.

I strongly value the mathematical development presented by the authors. The paper displays mastery of the theoretical application of DEA and SFA and how they can be used to evaluate options and make environmentally sustainable decisions.

Nonetheless, the paper is written in an unusual way for academic writing, making its readiness difficult. For instance, the introduction states the paper's goal but doesn't establish a context where this decision-making modeling can work. Later in the paper, one possible deduction is that the authors are thinking in public policy, but this is left to the readers to speculate.

Response:

Thank you for your insightful feedback on our manuscript. I have carefully reviewed your comments and made significant revisions to enhance the clarity and readiness of the paper.

Specifically, we have reworked the introduction to establish a clearer context for the proposed decision-making modeling within the realm of public policy. I have cited concrete examples and scenarios where the model could be effectively applied, particularly in the context of policymaking processes related to the bioeconomy. Additionally, I have explicitly elucidated the connection between the theoretical model and its practical implications for public policy, ensuring that readers can readily understand the relevance and significance of our work within this domain.

I believe that these revisions have substantially improved the readability and accessibility of the paper, addressing your concerns regarding the clarity of the introduction and the establishment of a context for the decision-making modeling. We appreciate your valuable feedback, which has undoubtedly strengthened the quality of our manuscript.

 [2] Also, in the introduction, some bullet ideas mention critical elements from the literature review, but these elements are not further developed in the paper. Later, there is a list of the paper's contributions; however, they are not linked with other sections of the introduction or literature about the topic.

Response:

Thank you for this observation. The bullets ideas indicated in the introduction moved to the literature review section. I consider that their observation in the introduction should be focused on the contribution of this study, highlighting that the model could support decision making. to decision makers and the public agenda. 

[3] Literature review: This section requires a profound review. Some constitutive elements of the concept are presented, but this is not a literature review. Here, as a reader, I expect to get information about the recent literature on bioeconomy, its applications, theoretical models, measurements, and so forth. In other words, I would like to read a review that allows me to understand the state of knowledge in this field and how this paper fits into the current literature. Additionally, due to the declaration of importance of this theory, a comprehensive explanation of Georgescu-Roegen's framework is deeply missed.

Response:

Dear reviewer 1 thanks for your observation. Regarding your observation “recent literature on bioeconomy, its applications, theoretical models” it is added in the “Economic Theories Relevant to the Bioeconomy”, “The measurement scopes of DEA-SFA for Georgescu-Roegen's bioeconomic model”, “Assumptions of the DEA Model - Georgescu-Roegen's Bioeconomic Model”, regarding to a comprehensive explanation of Georgescu-Roegen's framework is deeply missed. It is added in Results and Discussion section. “Georgescu-Roegen's bioeconomic model is characterized by several key principles, as outlined in the abstracts”. See it, after the Figure 3.

[4] Variables: This section presents methodological elements that must be considered when conducting such an endeavor, but it does not explain the chosen variables and the theoretical intuitions behind them.

Response:

Dear reviewer 1, thanks for this observation. Were added four paragraph for clarify it. 

[5] The following sections are challenging to follow up because it is not clear its intention to develop the paper's central argument. For example, the section "Economic theories relevant to the bioeconomy" is based on one author, and it presents very well-known ideas regarding the evolution of mathematical tools in economics. It is not clear why this information would be important in this context.

Response:

Thank you for this observation. Overall, the entire document has been reviewed to ensure coherence based on the proposed research objective. Well, the author, Jiménez (2014), provides an overview of the issue surrounding economic models. The third phase, from 1960 to the present day, highlights the development of dynamic analysis and the use of stochastic processes, employed by the new classical economics and the new Keynesian economics. So it is a introduction to the DEA and SFA incorporate to the model Georgescu Rogen. In this section was added a paragraph. 

[6] Finally, I strongly suggest a Discussion section. It is necessary to explain what these findings mean and how they contribute to the most recent literature. It is unclear if this model was thought for companies, countries, regions, or other subjects. Furthermore, an exploration of the model's limitations is required.

Dear reviewer 1, thanks for this observation, it is considering as Results and Discussion, where I include the most recent literature. Was added a paragraph for giving the context that you refer. Was added in the New Bioeconomy Model. 

Regarding to the unclear model was thought……..it is considered in the concluding remarks. A paragraph was added. 

Reviewer # 2

[7] Reviewer #2: This study aims to elucidate the bioeconomy's dynamic nature, constructing a comprehensive theoretical model.

It is quite interesting, with well supported data. However, there are quite many questions about the study.

Response:

Dear reviewer 2, thanks for your observation. In the conclusions section, we highlight the characteristics and gaps that any model may present. We consider that the contribution of this study encourages the challenge of identifying ways and techniques to refine the model in the socio-economic conditions of production units.

Reviewer # 3

[8] Reviewer #3: This is a confusing paper describing production functions and approaches to estimate productivity, claiming that would be a model of the Bioeconomy. It took me to read until the very end of the paper when the authors finally describe why they have presented all these equations. Most sections are completely unrelated to each other. Page numbers are missing (neither is there line numbering), making it really difficult to review. No numbering of sections, many headers the same size and font. Unclear how the paper is structured. Zero innovative content, not a scientific paper.

Response:

Dear reviewer 3 thanks for your observations. We are revised and order all the section to give the justification of each. We have put the numbers and the line numbering. Also we have put numbering to each section. Our contribution is the construction a Bioeconomy model as you can see in the literature reviewed there is a gap that we consider to fill. 

[9] More detailed comments:

Introduction

- Strange spacing, paragraphs after single sentences

Response:

It was corrected. 

- Strange structure, paragraphs starting with bullet points

Response:

It was corrected. 

- Typo: “the review literature“

- The following paragraph is a complete repetition of the previous sentences and redundant: “In conclusion, the bioeconomy does not have a single model, but rather encompasses diverse approaches and perspectives. It is important to consider the evolution of the concept, the crosssectional nature of the sector, and the shift towards a broader perspective for a comprehensive understanding of the bioeconomy [7, 8 9].”

Response:

It was corrected. It was eliminated.

- More one to one repetitions: “The bioeconomy has been framed differently across various fields and sectors”; followed by more repetitions of the exact same sentences

Response:

It was corrected. 

- The introduction is a repetition of the same arguments and the same references over and over again, using almost the exact same words, completely blown up without much content.

Response:

It was corrected. 

[10] Literature review

- Figures 1 and 2 are mixed up. The first part of the lit review seems to talk about figure 2 although there is no 1-to-1 overlap of text and figure. Both figures have no real sources. Figure 1 states the source: Scopus AI data. Unclear what that is.

- Shortest literature review ever

Response:

It was corrected. Source: Scopus AI data was eliminate. Both Figures refers the key word of the Bioeconomy and the second figure refer a map of this concept. 

[11] Variables

- Under the header “variables” comes a definition of different parts of models. There are no sources provided and it looks like it comes directly out of ChatGPT, because it is written by directly addressing the reader and giving a cooking recipe.

Response:

Thank you for this observation, the researchers' contributions on this topic have been better written, if the corresponding citations were added.

[12] - Economic Theories Relevant to the Bioeconomy: Suddenly change of citation style and strange paragraph on the history of economic models based on only one source

Response:

Thank you for noticing this information. Of course, in this section we wanted to start with this author's comment on how mathematically economic processes have been interpreted, realizing that in our time we are working with analysis of enveloping data. This allows us to justify in a certain way our Bioeconomy model based on stochastic processes to measure efficiency and productivity.

[13] - Assumptions of the DEA Model - Georgescu-Roegen's Bioeconomic Model: unclear listing of assumptions of DEA (Data Envelopment Analysis) and SFA (Stochastic Frontier Analysis) without the provision of sources

Response:

Dear reviewer, please check that is referenced with {37,45, 64], and remember that our model follow the consideration by author indicated in the review literature, specific Variable section.

[18]- The measurement scopes of DEA-SFA for Georgescu-Roegen's bioeconomic model: now comes a definition of DEA (Data Envelopment Analysis) and SFA (Stochastic Frontier Analysis); apparently there should some productivity measurement, but I only try to read between the lines

Response:

Dear reviewer, thanks for this observation. Of course this is correct. It was developed in the results section. 

[14]- Results and discussion: strange title, completely unrelated to text, talks about Georgescu-Roegen's model of Bioeconomy, but means “bioeconomics”; some other models and concepts are mentioned without explaining them. 

Response:

Dear reviewer, thanks for this observation, it was corrected. And was subdivided in Results Other section Discussion. 

[20]- Again figures 3 and 4 with source: Scopus AI data. Unclear what that is. Figures not really explained, but also seem trivial

Response:

Dear reviewer, thanks for this observation, it was corrected

[15]- A function is shown for Georgescu-Roegen’s Bioeconomic Model but not really explained, sources are in Spanish

Response:

Dear reviewer, thank you for your valuable comments, in this section it is rather to briefly present Georgescu Rogen's model, because in the review of the literature it has been explained, but rather to focus on how, from the model, we extend the proposed model as a result of our research.

Dear reviewer the [31] references was added. 

[16]- Then comes a description of The Constant Returns to Scale Model (CRS), DEA, some linear programming but it is all completely unclear how that relates to a comprehensive model for the bioeconomy 

 - 1.5) Panel Data, DEA and the Malmquist Index: more ways to measure productivity

 - SFA- Georgescu-Roegen's bioeconomic model: explaining a production function

Response:

Yes, my dear reviewer, in this section we explain the bioeconomy model with DEA and SFA methodology added.

[17]- “They are essential for the functioning of the economy but aren't always directly accounted for in traditional economic models.” Exact same sentence twice

[26]- Some other production function approaches are listed

Response:

Dear reviewer, it was corrected. 

[18]- Bioeconomy model (Equation 17): this is the most interesting section in the whole paper because it finally explains “DEA (Data Envelopment Analysis) and SFA (Stochastic Frontier Analysis) have potential applications in optimizing resource allocation in the bioeconomy.” And what the use of all the presentation of production function was. But there is no application at all. Just some arguments from the literature.

Response:

Dear reviewer, thanks for this comment. Well, this is a different moments of applications, it was discussing in the Discuss Section. However, in DEA and SFA are included the production function that you comment. 

[19] Concluding remarks

- Completely unrelated to the whole paper.

Response:

Dear reviewer, it was corrected.

---

## [Decision Letter · Decision Letter 1]

24 May 2024

PONE-D-23-43140R1Formulation of an Innovative Model for the Bioeconomy: Unraveling the Secrets of a Sustainable FuturePLOS ONE

Dear Dr. Zúniga-González,

Thank you for submitting your manuscript to PLOS ONE. After careful consideration, we feel that it has merit but does not fully meet PLOS ONE’s publication criteria as it currently stands. Therefore, we invite you to submit a revised version of the manuscript that addresses the points raised during the review process.

We look forward to receiving your revised manuscript.

Kind regards,

Noé Aguilar-Rivera

Academic Editor

PLOS ONE

Journal Requirements:

Reviewers' comments:

Reviewer's Responses to Questions

**Comments to the Author**

1. If the authors have adequately addressed your comments raised in a previous round of review and you feel that this manuscript is now acceptable for publication, you may indicate that here to bypass the “Comments to the Author” section, enter your conflict of interest statement in the “Confidential to Editor” section, and submit your "Accept" recommendation.

Reviewer #4: (No Response)

Reviewer #5: All comments have been addressed

Reviewer #6: All comments have been addressed

2. Is the manuscript technically sound, and do the data support the conclusions?

Reviewer #4: Partly

Reviewer #5: Yes

Reviewer #6: Yes

3. Has the statistical analysis been performed appropriately and rigorously? 

Reviewer #4: No

Reviewer #5: Yes

Reviewer #6: I Don't Know

4. Have the authors made all data underlying the findings in their manuscript fully available?

Reviewer #4: (No Response)

Reviewer #5: Yes

Reviewer #6: Yes

5. Is the manuscript presented in an intelligible fashion and written in standard English?

Reviewer #4: (No Response)

Reviewer #5: Yes

Reviewer #6: No

6. Review Comments to the Author

**Reviewer #4:** May 13, 2024

PLOS ONE

Reviewer Recommendation and Comments for Manuscript Number PONE-D-23-43140R1

Title: Formulation of an Innovative Model for the Bioeconomy: Unraveling the Secrets of a

Sustainable Future

Reviewer Comments to Author

Review summary: As proposed in the Abstract, the authors aim to “elucidate the bioeconomy's dynamic nature” and construct a comprehensive theoretical model by conducting an “extensive literature review.” Their results incorporate “Georgescu-Roegen's insights.” As such the Abstract is well written. However, the remainder of the manuscript does not follow the same pattern as outlined in the abstract; therefore, it is difficult to follow the entire manuscript. The subtitle (Unraveling the Secrets of a Sustainable Future) that follows the colon in Title has not really been demonstrated in the manuscript.

The Introduction section attempts to define the term bioeconomy, but it fails to contextualize with their effort that specially incorporates “Georgescu-Roegen's insights” in the Introduction. Before publishing this manuscript, I recommend illustrating the Equations (1) and (2) using some theoretical data. I also recommend presenting the manuscript content as they have hypothesized in the Abstract. Additional suggestions are offered below:

1. Line 95: “Ultimately, this study aims to contribute to redefining the bioeconomy to make it a more cohesive and sustainable by providing a unifying framework that takes into accommodates its diverse facets in a timeline and fosters collaboration and informed decision-making.” This statement needs revision.

2. The font and font size are inconsistent in Sections 2 and 2.1.

3. The sentences in Line 224-226 and Line 229-230 should be revised.

4. Line 284 citation needs to be updated.

5. The statement in Line 149-165 emphasizes that “The bioeconomy revolves around the use of resources that can be naturally replenished, such as crops, forests, and microorganisms ……….. It seeks to minimize the environmental impact of resource utilization, reduce greenhouse gas emissions, and promote responsible land and resource management.” It would be meaningful to contrast other similar concepts and methods that utilize Data Envelopment Analysis (DEA) with what the authors have described. For example, the concept of eco-efficiency and sustainability.

6. The abbreviations must be spelled out at the first appearance, see Line 321 CRS.

7. Equation 1 and Equation 2 in Page 18 are unclear. The best way to describe these equations would be to illustrate the Q0(t) using some theoretical data.

8. The literature review is incomplete. Although they have attempted to provide some definitions of bioeconomy, they fail to conduct a systematic literature review. Suggested example literature on concept of bioeconomy: https://doi.org/10.23987/sts.69662.

9. The statement in Concluding Remark section (Line 717-718) could be supported with the illustration of DEA using some theoretical data if realistic data is not available.

10. The purpose of mathematical models should be clarified using example data.

For all the above reasons, I recommend Major Revision.

**Reviewer #5: **The authors have incorporated most of the suggestions made by the Reviewers and therefore, the MS may be accepted for publication in PLoS ONE.

**Reviewer #6: **This paper has carried out a deep and interesting exploration of concept and mathematical methods for "Model for the Bioeconomy", which has a good innovation. In general, the author made corresponding modifications according to the opinions of the two reviewers in the first round, and the quality of the paper was improved. But there are still some problems worth further improvement.

1) In the title, "Unraveling the Secrets of a Sustainable Future" exaggerates the significance of the content of the paper, and it is suggested that this part should be deleted or properly adjusted.

2) There are too many paragraphs in many parts of the full text, so it is necessary to summarize the main arguments into a few paragraphs to enhance the readability of the article.

3) The aesthetics and logic of the concept figure need to be strengthened, and the relationships in Figure 1 are confusing.

4) The formula expression in the paper should be modified according to the standard, and many formulas lack necessary symbols, such as brackets.

5) The format of the full text is confused, and the title of each level is not uniform, which needs to be adjusted and standardized according to the requirements of the journal.

6) The discussion part should focus on the innovation and practical application of the new model constructed by the author.

7) What is the role of "Concluding remarks"? It is suggested that many contents should be included in the introduction as an explanation of "knowledge gap" to highlight the innovation of this research.

7. PLOS authors have the option to publish the peer review history of their article (what does this mean?). If published, this will include your full peer review and any attached files.

Reviewer #4: No

Reviewer #5: No

Reviewer #6: No

---

## [Author Response · Author response to Decision Letter 1]

7 Jun 2024

Noé Aguilar-Rivera

Editor Academico

Dear editor, thank you for all your observations, in this third round, that surely contribute to improving the quality of the research. Below I present in detail each of the improvements incorporated.

Reviewers' comments:

Reviewer #4: May 13, 2024

PLOS ONE

Reviewer Recommendation and Comments for Manuscript Number PONE-D-23-43140R1

Title: Formulation of an Innovative Model for the Bioeconomy: Unraveling the Secrets of a Sustainable Future

Reviewer Comments to Author

[1] Review summary: As proposed in the Abstract, the authors aim to “elucidate the bioeconomy's dynamic nature” and construct a comprehensive theoretical model by conducting an “extensive literature review.” Their results incorporate “Georgescu-Roegen's insights.” As such the Abstract is well written. However, the remainder of the manuscript does not follow the same pattern as outlined in the abstract; therefore, it is difficult to follow the entire manuscript. The subtitle (Unraveling the Secrets of a Sustainable Future) that follows the colon in Title has not really been demonstrated in the manuscript.

[1] Response to reviewer 4:

Thanks reviewer 4 for your observations, abstract was enhance marked up in green color. 

Regarding to the “The subtitle (Unraveling the Secrets of a Sustainable Future) that follows the colon in Title has not really been demonstrated in the manuscript.” It was eliminate as was suggesting by reviewer 6 too. 

A thorough review of the manuscript was conducted for coherence based on what was outlined in the abstract. The improvements are included in green.

[2] The Introduction section attempts to define the term bioeconomy, but it fails to contextualize with their effort that specially incorporates “Georgescu-Roegen's insights” in the Introduction. 

Response to the reviewer 4

Thanks for this observation, was added a paragraph in line 87-91. 

Before publishing this manuscript, I recommend illustrating the Equations (1) and (2) using some theoretical data. 

Dear author thanks for this observation, the ec. 17 was improvement adding the CCR-CRS and BCC-VRS model, and the adding the Figure 5, because not only is it about illustrating equations 1 and 2, but the challenge lies in integrating the 13 equations with the DEA and SFA methodology, which are the most used methods for measuring productivity and efficiency in recent decades, and integrating them into the model. Please check the figure 5, and thanks because your observation help us to improvement the equation 17. 

Response to the reviewer 4

I also recommend presenting the manuscript content as they have hypothesized in the Abstract. 

Response to the reviewer 4

Dear reviewer 4 thanks for this observation, it was added in the line 73-78, 83-88 y 98-99

Additional suggestions are offered below:

1. Line 95: “Ultimately, this study aims to contribute to redefining the bioeconomy to make it a more cohesive and sustainable by providing a unifying framework that takes into accommodates its diverse facets in a timeline and fosters collaboration and informed decision-making.” This statement needs revision.

Response to reviewer 4:

Thanks for your observation, it was improvement: “Ultimately, this study aims to contribute to redefining the bioeconomy to make it a more cohesive and sustainable by providing a unifying framework that takes into accommodates its diverse facets over time, fostering collaboration and informed decision-making”.

2. The font and font size are inconsistent in Sections 2 and 2.1.

Response to reviewer 4:

It was corrected.

3. The sentences in Line 224-226 and Line 229-230 should be revised.

Response to reviewer 4:

Dear reviewer it was improvement: Throughout the manuscript, it systematically reference the contributions of various authors who have reevaluated and reimagined the bioeconomy model proposed, thereby enriching its holistic integration [58, 61, 66, 68]. These contributions are pivotal in refining our understanding and application of the bioeconomy framework, ensuring its relevance and effectiveness in addressing contemporary challenges. Specifically, we have reconsidered the foundational principles outlined by these authors, integrating their insights to enhance the comprehensiveness and applicability of our proposed model. By acknowledging and incorporating these diverse perspectives, we strive to develop a more robust and inclusive framework for navigating the complexities of the bioeconomy landscape.

4. Line 284 citation needs to be updated.

Response to reviewer 4:

Dear reviewer 4, thanks for this observation, the cite 45 was update with 75.

5. The statement in Line 149-165 emphasizes that “The bioeconomy revolves around the use of resources that can be naturally replenished, such as crops, forests, and microorganisms ……….. It seeks to minimize the environmental impact of resource utilization, reduce greenhouse gas emissions, and promote responsible land and resource management.” It would be meaningful to contrast other similar concepts and methods that utilize Data Envelopment Analysis (DEA) with what the authors have described. For example, the concept of eco-efficiency and sustainability.

Response to reviewer 4:

Thank you for your insightful observation. In response to your suggestion, we have same idea about the comparison of the bioeconomy with similar concepts such as eco-efficiency and sustainability, particularly in the context of Data Envelopment Analysis (DEA). We add this paragraph 

“In summary, while the bioeconomy, eco-efficiency, and sustainability share the goal of reducing environmental impact and improving efficiency, each approaches this goal from different perspectives and with distinct focuses. DEA is a versatile tool that can be adapted to evaluate efficiency in each of these contexts, providing a quantitative perspective on performance based on resources used and outcomes achieved”.

We appreciate your valuable feedback and believe this addition enhances the clarity and depth of our manuscript.

6. The abbreviations must be spelled out at the first appearance, see Line 321 CRS.

Response to reviewer 4:

Dear reviewer it was added:” (Standard Constant Returns to Scale (CRS) and Variable Returns to Scale (VRS) models)”. 

7. Equation 1 and Equation 2 in Page 18 are unclear. The best way to describe these equations would be to illustrate the Q0(t) using some theoretical data.

Response to reviewer 4:

8. The literature review is incomplete. Although they have attempted to provide some definitions of bioeconomy, they fail to conduct a systematic literature review. Suggested example literature on concept of bioeconomy: https://doi.org/10.23987/sts.69662.

Response to reviewer 4:

The cite 76 was added, thanks for this observation. 

9. The statement in Concluding Remark section (Line 717-718) could be supported with the illustration of DEA using some theoretical data if realistic data is not available.

Response to reviewer 4:

Thank you for your suggestion. In response to your feedback, we have added an illustration of DEA using theoretical data to support the statement in the Concluding Remark section (Lines 717-718). This information has been included in the Supporting Information (S1) section.

10. The purpose of mathematical models should be clarified using example data.

Response to reviewer 4:

Thank you for your suggestion. To clarify the purpose of the mathematical models, we have expanded the illustration in the Supporting Information (S2) to include a detailed explanation of how Data Envelopment Analysis (DEA) and Stochastic Frontier Analysis (SFA) can be applied using example data. This expanded illustration demonstrates the practical application and benefits of these models in optimizing resource allocation and improving efficiency in the bioeconomy.

For all the above reasons, I recommend Major Revision.

Reviewer #5: The authors have incorporated most of the suggestions made by the Reviewers and therefore, the MS may be accepted for publication in PLoS ONE

Response to reviewer 5

Thanks for your observation. 

Reviewer #6: This paper has carried out a deep and interesting exploration of concept and mathematical methods for "Model for the Bioeconomy", which has a good innovation. In general, the author made corresponding modifications according to the opinions of the two reviewers in the first round, and the quality of the paper was improved. But there are still some problems worth further improvement.

1) In the title, "Unraveling the Secrets of a Sustainable Future" exaggerates the significance of the content of the paper, and it is suggested that this part should be deleted or properly adjusted.

Response to Reviewer 6

Thanks for your observations, the authors following your suggest and was deleted. 

2) There are too many paragraphs in many parts of the full text, so it is necessary to summarize the main arguments into a few paragraphs to enhance the readability of the article.

Response to Reviewer 6

I would like to sincerely thank you for your detailed review of our document. We have taken into account your observations and have conducted a general revision of the document to enhance its readability and clarity. Summaries of the main parts have been provided, which we hope will make the content more accessible and understandable for readers.

If you have any further suggestions or comments, please do not hesitate to let us know. We greatly appreciate your time and dedication to improving the quality of our work.

3) The aesthetics and logic of the concept figure need to be strengthened, and the relationships in Figure 1 are confusing.

Response to Reviewer 6

Thank you for your valuable feedback. We have revised and improved the aesthetics and logical clarity of Figure 1 as per your suggestion. The relationships have been reorganized to ensure they are more intuitive and easier to understand.

The updated Figure 1 now clearly illustrates the components and their relationships within the bioeconomy model, differentiating between the inputs and outputs with distinct visual elements.

4) The formula expression in the paper should be modified according to the standard, and many formulas lack necessary symbols, such as brackets.

Response to Reviewer 6

Thank you for your valuable feedback. We have carefully reviewed the expressions and formulas in the paper according to the standard conventions, ensuring the inclusion of necessary symbols such as brackets. We have made the necessary modifications to enhance clarity and precision in the mathematical expressions presented in the document.

5) The format of the full text is confused, and the title of each level is not uniform, which needs to be adjusted and standardized according to the requirements of the journal.

Response to Reviewer 6

Thank you for your review and constructive feedback on the formatting of our manuscript. We have carefully addressed your comments and made necessary adjustments to improve the clarity and uniformity of the text, particularly regarding the presentation of formulas. Here's how we've addressed your concerns:

We have thoroughly revised the presentation of formulas throughout the manuscript to ensure clarity and adherence to standard formatting conventions. Each formula is now presented in a clear and uniform manner, with all necessary symbols and notation included for better comprehension.

We have also reviewed the overall formatting of the manuscript to ensure uniformity and consistency in the title hierarchy. The titles of each section and subsection have been adjusted and standardized according to the requirements of the journal, enhancing the overall readability and organization of the text.

We believe that these improvements have significantly enhanced the presentation of the manuscript and addressed the formatting issues you raised. We appreciate your attention to detail and your commitment to maintaining the quality standards of the journal.

6) The discussion part should focus on the innovation and practical application of the new model constructed by the author.

Response to Reviewer 6

Dear Reviewer, Thank you for your insightful feedback. We have revised the discussion section to focus on the innovation and practical application of the new model constructed in the paper. The revised discussion now highlights the potential applications of DEA and SFA methodologies in optimizing resource allocation within the bioeconomy, as well as their integration into our newly constructed model.

We have emphasized the innovative aspects of our approach, particularly in addressing inefficiencies and optimizing resource allocation within the bioeconomy. Additionally, we have provided a comprehensive comparison of various bioeconomy models, including Georgescu-Roegen's foundational model, to elucidate their advantages, limitations, and applicability in diverse economic and environmental contexts.

Furthermore, we have explored how ethical and environmental considerations are integrated into the analysis of efficiency and productivity using DEA and SFA, aligning with Georgescu-Roegen's emphasis on sustainability and resource conservation.

Lastly, we have addressed the challenges in transitioning to a more sustainable bioeconomy and proposed strategies to overcome these challenges, underscoring the importance of effective public policies in fostering resilience and equity.

We believe these revisions enhance the discussion section, providing readers with valuable insights into the practical implications and innovations of our new model.

7) What is the role of "Concluding remarks"? It is suggested that many contents should be included in the introduction as an explanation of "knowledge gap" to highlight the innovation of this research.

Response to Reviewer 6

Thank you for your thorough review and valuable feedback on our manuscript. We have carefully considered each of your suggestions and made significant revisions to improve the clarity, coherence, and focus of the paper. Here are our responses to your comments:

Introduction Enhancement: We have integrated a clearer explanation of the research's innovation and contribution to addressing the knowledge gap into the introduction. This addition highlights the significance of our study in advancing understanding and decision-making within the bioeconomy context.

Discussion Focus on Innovation and Practical Application: We have revised the discussion section to focus more explicitly on the innovation and practical application of the new model constructed in this study. By highlighting the innovative aspects and potential real-world applications of our model, we aim to provide a clearer understanding of its relevance and utility for decision-makers.

Concluding Remarks Role Clarification: We understand your suggestion regarding the role of "Concluding Remarks" and will revise this section accordingly. We will include relevant contents from the conclusion into the introduction, emphasizing the identification of the knowledge gap and the innovative aspects of our research.

Formula Expression Standardization: We have carefully reviewed the formulas in the paper and modified them according to standard formatting conventions. We have also ensured that all necessary symbols, such as brackets, are included for clarity and consistency.

Discussion Content on DEA and SFA Applications: The discussion section now provides a more detailed exploration of the potential applications of Data Envelopment Analysis (DEA) and Stochastic Frontier Analysis (SFA) in optimizing resource allocation within the bioeconomy. We have highlighted key points debatable to support this claim, including the role of DEA and SFA in uncovering inefficiencies and driving eco-efficiency.

We believe that these revisions have significantly strengthened the manuscript and addressed your concerns effectively. We appreciate your thoughtful feedback, which has undoubtedly improved the quality and impact of our research. If you have any further suggestions or require additional clarification, please don't hesitate to let us know.

---

## [Decision Letter · Decision Letter 2]

12 Jul 2024

PONE-D-23-43140R2Formulation of an Innovative Model for the BioeconomyPLOS ONE

Dear Dr. Zúniga-González,

Thank you for submitting your manuscript to PLOS ONE. After careful consideration, we feel that it has merit but does not fully meet PLOS ONE’s publication criteria as it currently stands. Therefore, we invite you to submit a revised version of the manuscript that addresses the points raised during the review process.

We look forward to receiving your revised manuscript.

Kind regards,

Noé Aguilar-Rivera

Academic Editor

PLOS ONE

Journal Requirements:

Reviewers' comments:

Reviewer's Responses to Questions

**Comments to the Author**

1. If the authors have adequately addressed your comments raised in a previous round of review and you feel that this manuscript is now acceptable for publication, you may indicate that here to bypass the “Comments to the Author” section, enter your conflict of interest statement in the “Confidential to Editor” section, and submit your "Accept" recommendation.

Reviewer #6: All comments have been addressed

Reviewer #7: (No Response)

2. Is the manuscript technically sound, and do the data support the conclusions?

Reviewer #6: Yes

Reviewer #7: Yes

3. Has the statistical analysis been performed appropriately and rigorously? 

Reviewer #6: (No Response)

Reviewer #7: Yes

4. Have the authors made all data underlying the findings in their manuscript fully available?

Reviewer #6: (No Response)

Reviewer #7: Yes

5. Is the manuscript presented in an intelligible fashion and written in standard English?

Reviewer #6: (No Response)

Reviewer #7: Yes

6. Review Comments to the Author

Reviewer #6: The author has made substantial replies and modifications to the questions I raised in the previous round, and the quality of the paper has been improved. After further improvement of some language and format problems, it is recommended that this paper be accepted and published.

Reviewer #7: The manuscript is very interesting, it addresses a key issue in the difficulty of bioeconomic studies. The models studied are statistically strong for the topic of economics, however, for the social topic they can be subjective. The discussion could be strengthened by including literature on impact measurements in bioeconomy and policies for the growth of the bioeconomy, but in emerging countries, analyze if they exist, how they have impacted and if they favor sustainable development indices.

Spelling errors are observed in the figures of the manuscript.

7. PLOS authors have the option to publish the peer review history of their article (what does this mean?). If published, this will include your full peer review and any attached files.

Reviewer #6: No

Reviewer #7: **Yes: **LUIS ALBERTO OLVERA VARGAS

---

## [Author Response · Author response to Decision Letter 2]

15 Jul 2024

Reviewers' comments:

Reviewer's Responses to Questions

Comments to the Author

Reviewer #6: The author has made substantial replies and modifications to the questions I raised in the previous round, and the quality of the paper has been improved. After further improvement of some language and format problems, it is recommended that this paper be accepted and published.

Response: Dear reviewer 6 thanks for your contribution. 

Reviewer #7: The manuscript is very interesting, it addresses a key issue in the difficulty of bioeconomic studies. The models studied are statistically strong for the topic of economics, however, for the social topic they can be subjective. The discussion could be strengthened by including literature on impact measurements in bioeconomy and policies for the growth of the bioeconomy, but in emerging countries, analyze if they exist, how they have impacted and if they favor sustainable development indices.

Spelling errors are observed in the figures of the manuscript.

Response: Reviewer #7's insightful feedback is greatly appreciated. They find the manuscript on bioeconomic studies addressing a key issue, with statistically strong models in economics but potentially subjective in social aspects. The reviewer suggests strengthening the discussion with literature on impact measurements in bioeconomy and policies for its growth in emerging countries, emphasizing their impact on sustainable development indices. Regarding the applicability to Georgescu-Roegen's bioeconomic model, while foundational in ecological economics, it primarily focuses on theoretical rather than specific policy or measurement frameworks for emerging economies. To address this, the manuscript will bolster its discussion with empirical evidence and case studies exploring the implementation and socio-economic impacts of bioeconomic policies in diverse contexts.

The Figure 5 and 6 were improvement by the spelling errors, we give the thanks to the reviewer for this. 

Bst Rgs

Carlos

---

## [Decision Letter · Decision Letter 3]

12 Aug 2024

Formulation of an Innovative Model for the Bioeconomy

PONE-D-23-43140R3

Dear Dr. C. A. Zúniga-González

We’re pleased to inform you that your manuscript has been judged scientifically suitable for publication and will be formally accepted for publication once it meets all outstanding technical requirements.

Kind regards,

Noé Aguilar-Rivera

Academic Editor

PLOS ONE

---

## [Editor Report · Acceptance letter]

16 Aug 2024

PONE-D-23-43140R3 

PLOS ONE

Dear Dr. Zúniga-González, 

I'm pleased to inform you that your manuscript has been deemed suitable for publication in PLOS ONE. Congratulations! Your manuscript is now being handed over to our production team.

Kind regards, 

on behalf of

Dr. Noé Aguilar-Rivera 

Academic Editor

PLOS ONE